ⓐ | **Open Peer Review** | Human Microbiome | Research Article

# Systematic identification of secondary bile acid production genes in global microbiome

Yuwei Yang,[1] Wenxing Gao,[1] Ruixin Zhu,[1] Liwen Tao,[1] Wanning Chen,[1] Xinyue Zhu,[1] Mengping Shen,[1] Tingjun Xu,[1,2] Tingting Zhao,[1,3] Xiaobai Zhang,[1] Lixin Zhu,[4] Na Jiao[5]

**ABSTRACT** Microbial metabolism of bile acids (BAs) is crucial for maintaining homeostasis in vertebrate hosts and environments. Although certain organisms involved in bile acid metabolism have been identified, a global, comprehensive elucidation of the microbes, metabolic enzymes, and bile acid remains incomplete. To bridge this gap, we employed hidden Markov models to systematically search in a large-scale and high-quality search library comprising 28,813 RefSeq multi-kingdom microbial complete genomes, enabling us to construct a secondary bile acid production gene catalog. This catalog greatly expanded the distribution of secondary bile acid production genes across 11 phyla, encompassing bacteria, archaea, and fungi, and extended to 14 habitats spanning hosts and environmental contexts. Furthermore, we highlighted the associations between secondary bile acids (SBAs) and gastrointestinal and hepatic disorders, including inflammatory bowel disease (IBD), colorectal cancer (CRC), and nonalcoholic fatty liver disease (NAFLD), further elucidating disease-specific alterations in secondary bile acid production genes. Additionally, we proposed the pig as a particularly suitable animal model for investigating secondary bile acid production in humans, given its closely aligned secondary bile acid production gene composition. This gene catalog provides a comprehensive and reliable foundation for future studies on microbial bile acid metabolism, offering new insights into the microbial contributions to health and disease.

**IMPORTANCE** Bile acid metabolism is an important function in both host and environmental microorganisms. The existing functional annotations from single source pose limitations on cross-habitat analysis. Our construction of a systematic secondary bile acid production gene catalog encompassing numerous high-quality reference sequences propelled research on bile acid metabolism in the global microbiome, holding significance for the concept of One Health. We further highlighted the potential of the microbiota-secondary bile acid axis as a target for the treatment of hepatic and intestinal diseases, as well as the varying feasibility of using animal models for studying human bile acid metabolism. This gene catalog offers a solid groundwork for investigating microbial bile acid metabolism across different compartments, including humans, animals, plants, and environments, shedding light on the contributions of microorganisms to One Health.

**KEYWORDS** secondary bile acid, microbiome, gene catalog, microbial metabolism, gastrointestinal and hepatic pathophysiology

Microbial-derived secondary bile acids (SBAs) exert multifaceted influence on vertebrate hosts through various mechanisms. These mechanisms include direct cytotoxicity (1), direct DNA damage (2), and activation of receptors distributed across multiple tissues, including the liver, intestine, brain, and breast (3–5). Secondary bile acids originate from intestinal (6) microbial enzymatic transformations on primary bile

Address correspondence to Ruixin Zhu, rxzhu@tongji.edu.cn, Lixin Zhu, zhulx6@mail.sysu.edu.cn, or Na Jiao, najiao@fudan.edu.cn.

Yuwei Yang and Wenxing Gao contributed equally to this article. Author order was determined by drawing straws.

The authors declare no conflict of interest.

See the funding table on p. 17.

acids (PBAs) synthesized from cholesterol by host liver (7). The principal microbial transformations of bile acids (BAs) comprise several key processes: deconjugation by bile salt hydrolases (BSHs) (8), dehydroxylation by proteins encoded by bile acid-inducible (Bai) genes (9–15), oxidation, and epimerization by position-specific α/β-hydroxysteroid dehydrogenases (α/β-HSDHs) (16).

Disruption of microbial bile acid metabolism affects bile acid production and transport (17), lipid and glucose metabolism (18), as well as innate and adaptive immunity (19), thereby contributing to the pathogenesis of a broad spectrum of diseases (20, 21). For instance, elevated deoxycholic acid (DCA) levels in the liver can facilitate hepatocellular carcinoma development by promoting the secretion of pro-inflammatory and tumor-promoting mediators (22). In digestive tract, DCA also exacerbates intestinal inflammation by upregulating hepatic *de novo* bile acid synthesis (23). Furthermore, certain animal-derived secondary bile acids, such as hyodeoxycholic acid in pigs and ursodeoxycholic acid (UDCA) in bears, serve as therapeutic agents for conditions like nonalcoholic fatty liver disease (NAFLD) (24) and fatal veno-occlusive disease (25), respectively. Besides their important roles in the host, approximately 5% of bile acids are released into the environment by vertebrate feces and urine (8, 26). Environmental microorganisms metabolize these bile acids as carbon- and energy-rich growth substrates and produce hormone-like metabolites, interfering with signaling systems to exert ecological effects (27). The isolation of environmental bile acid-metabolizing microorganisms (28, 29) reveals the ubiquity of microbial bile acid metabolism and highlights how bile acids and microorganisms from various hosts and environments can interconnect through food webs (30). Microorganisms form complex ecological relationships (31) and complement each other's bile acid pathways, broadening the bile acid production repertoire (32). Thus, both bottom-up control mediated by food and top-down control triggered by the impact of predation on prey in the ecosystem (33) have the potential to influence the cycle and diversity of microbial-derived bile acids, which are vital components of global health.

Several microorganisms capable of metabolizing bile acid have been reported, primarily including *Bifidobacterium* (34), *Enterococcus* (35), and *Listeria* (36) for bile acid deconjugation; *Clostridium* (37) for dehydroxylation; and *Eggerthella* (38) and *Ruminococcus* (16) for bile acid oxidation and epimerization. Early efforts to identify bile acid metabolism enzymes in microorganisms were largely dependent on labor-intensive biochemical methods like enzyme activity assay (39) and immunoblot analysis (40), which were limited in efficiency. Recent advancements in sequencing technologies and bioinformatics have revolutionized this field, enabling more rapid and accurate identification of bile acid metabolism genes. Computational tools like BLAST and MUSCLE have been successfully applied to identify specific bile acid-metabolizing enzymes within microbial genomes. For example, bile salt hydrolases were identified in human fecal metagenomic data sets (41), BaiE was characterized in metagenome-assembled genomes (MAGs) derived from human gastrointestinal tract and fecal data sets (42), and 7α/7β-HSDHs were discovered in black bear fecal metagenomic data sets (43). Heinken et al. expanded the scope of research by systematically identifying microbial bile acid deconjugation and biotransformation with MUSCLE across a curated set of 693 human gut microbial genomes (32). However, this study was limited by its relatively small genome set compared to the 3,594 high-quality species genomes included in human gut microbiome reference data set reported in 2022 (44). Despite these significant advances, much of the research has focused on isolated bile acid metabolic functions and narrow sets of microbial genomes from single-host system. This focus hinders a broader understanding of the intricate global ecological network underpinning microbial bile acid metabolism across diverse environments.

In response to these challenges, we aimed to expand the understanding of bile acid microbial metabolism in the global microbiome and explore its critical implications for One Health (45). To achieve this, we utilized a comprehensive database of 28,813 multi-kingdom microbial complete genomes from RefSeq and employed hidden

Markov models (HMMs) to construct a comprehensive gene catalog for secondary bile acid production. This catalog encompasses key genes from major metabolic pathways, including bile salt hydrolases, bile acid-inducible genes, and hydroxysteroid dehydrogenases. This function-rich catalog, derived from a diverse array of cultured microorganisms spanning various habitats, serves as a reliable reference for annotating secondary bile acid production genes in global metagenomic data sets. Additionally, this resource facilitates systematic exploration of bile acid metabolism across different species and microbial communities, offering new insights into the intricate relationships that govern these processes in diverse environments.

## RESULTS

### Global map of secondary bile acid-metabolic gene and microorganism

#### Overview of secondary bile acid production gene catalog

To construct a more comprehensive gene catalog associated with secondary bile acid metabolism, we developed HMMs for 13 gene families involved in this process, respectively. Using these HMMs, we performed a systematic search across 28,813 complete genomes of bacteria, archaea, and fungi in the RefSeq database. To ensure the accuracy and reliability of our findings, rigorous screening criteria were applied, based on the significance (e-value) and similarity (HMM score) of each hit, as detailed in the Methods section. The resulting secondary bile acid production gene catalog encompassed a total of 1,668 bile salt hydrolase genes, 241 bile acid-inducible genes, 159 3α-hydroxysteroid dehydrogenase genes, 136 3β-hydroxysteroid dehydrogenase genes, 2,770 7α-hydroxysteroid dehydrogenase genes, seven 7β-hydroxysteroid dehydrogenase genes, and 386 12α-hydroxysteroid dehydrogenase genes (Fig. S1; Fig. 1a).

Further analysis revealed variability in the copy number of these genes across different genomes (Fig. 1a). Single-copy genes were most common among 3αHSDH, 3βHSDH, and 7βHSDH, whereas, bile salt hydrolase, 7αHSDH, and 12αHSDH were often found in multi-copy within certain genomes. Notably, duplication of bile salt hydrolase gene was particularly prevalent, appearing in up to 271 genomes across 28 species of Firmicutes. In some cases, such as eight genomes from *Enterococcus faecium*, up to four copies of the bile salt hydrolase gene were identified. Additionally, multi-copy expression of bile salt hydrolase was a common feature in *E. faecium* genomes (Fig. 1b), with 77.3% (194 out of 251 genomes) exhibiting this trait. Regarding 7αHSDH, two copies were identified within Proteobacteria, specifically in one genome from *Acinetobacter* and three genomes from *Mesorhizobium* (Fig. 1c). For 12αHSDH, multi-copy genes were observed across three phyla: Actinobacteria, Firmicutes, and Proteobacteria. The genome of *Denitratisoma oestradiolicum* (GCF_902813185.1) within Proteobacteria harbored three copies, while 23 other genomes contained two copies (Fig. 1d). *Prescottella equi*, a member of Actinobacteria, was the primary source of genomes with two copies of 12αHSDH. These findings highlight the differences in gene copy number associated with secondary bile acid production across diverse microbial taxa. Such variability may enhance our understanding of the metabolic capabilities and ecological competitiveness of microorganisms involved in secondary bile acid metabolism.

#### Taxonomic distribution of secondary bile acid production genes

The broad coverage of microbial genomes and diverse metabolic genes allowed us to investigate the distribution of secondary bile acid production genes across a wide range of microbial species and even across different strains. By leveraging the lineage information from the RefSeq database, we calculated the numbers of bile acid metabolism genes across multiple taxonomic ranks to systematically explore the distribution characteristics.

Bile salt hydrolase genes were predominantly present in Firmicutes (1,374 genes, 82.4%) and Actinobacteria (248 genes, 14.9%). At the genus level, the majority of bile salt hydrolase genes within Firmicutes were distributed into genera, such as *Enterococcus* (655 genes, 39.3%), *Listeria* (281 genes, 16.8%), and *Lactiplantibacillus* (157 genes, 9.4%)

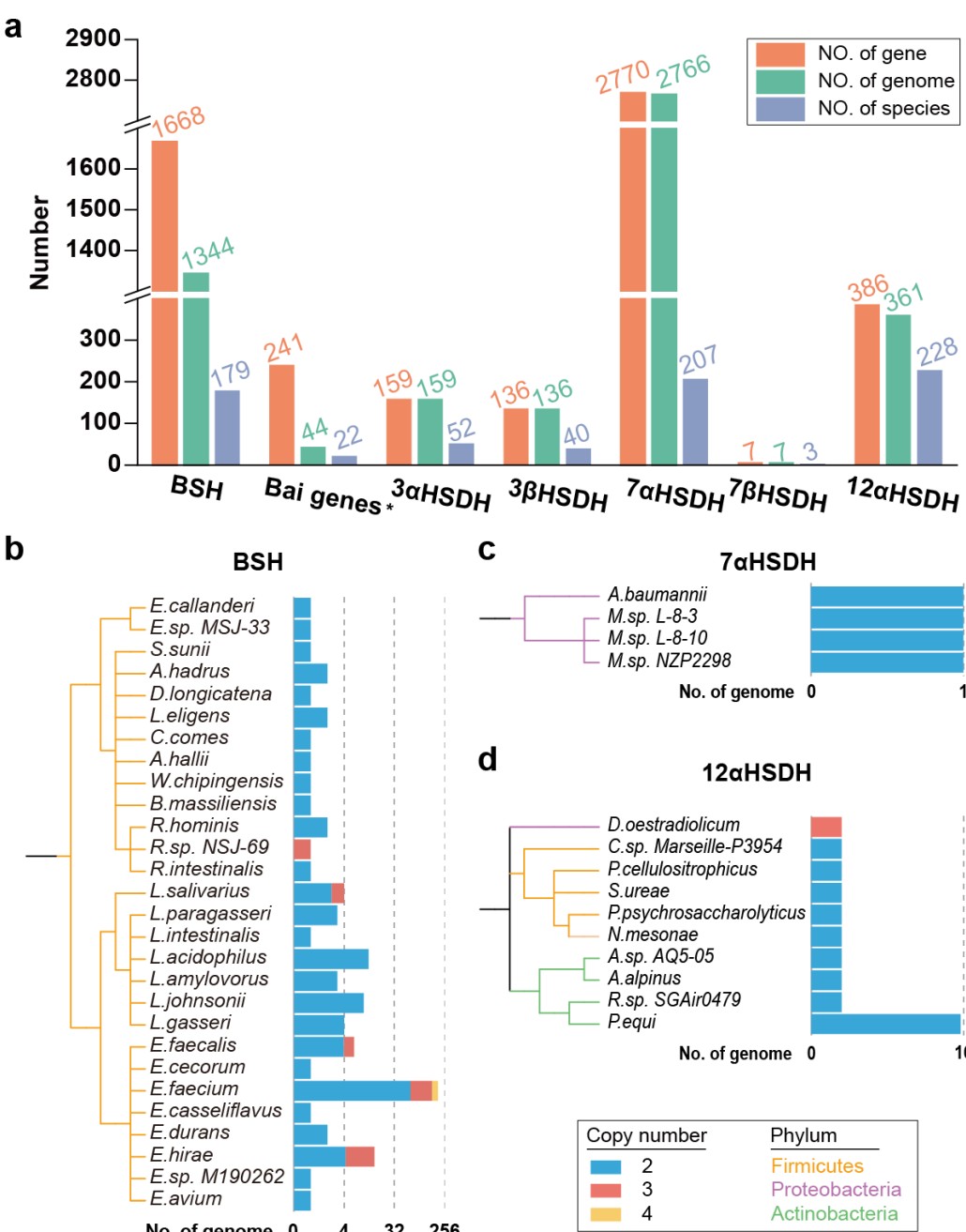

**FIG 1** Overview of secondary bile acid production gene catalog. (a) The number of secondary bile acid production genes (orange) as well as microbial genomes (green) and species (purple) carrying secondary bile acid production genes. Phylogenetic trees of species with genome carrying duplicate (b) bile salt hydrolase, (c) 12αHSDH, and (d) 7αHSDH genes. The branch colors represent different phyla. Stacked bar charts aligned to tree tips represent the number of genomes with duplicate genes. The blue, pink, and yellow bars indicate the number of genomes with different copy numbers.

(Fig. 2a). Notably, among these, 273 genes were identified in 98.9% (273 genomes) of *Listeria monocytogenes* genomes, and 156 genes were found in 87.6% (156 genomes) of *Lactiplantibacillus plantarum* genomes (Table S5), indicating that these species typically possess a single copy of bile salt hydrolase. Additionally, our findings also expanded the distribution of bile salt hydrolase in archaea beyond the previously reported species *Methanobrevibacter smithii* and *Methanosphaera stadtmanae* (32), bile salt hydrolase genes were also discovered in species such as *Methanobrevibacter millerae* and *Methanobrevibacter olleyae* of Euryarchaeota, as well as *Candidatus Methanomethylophilus alvus* of

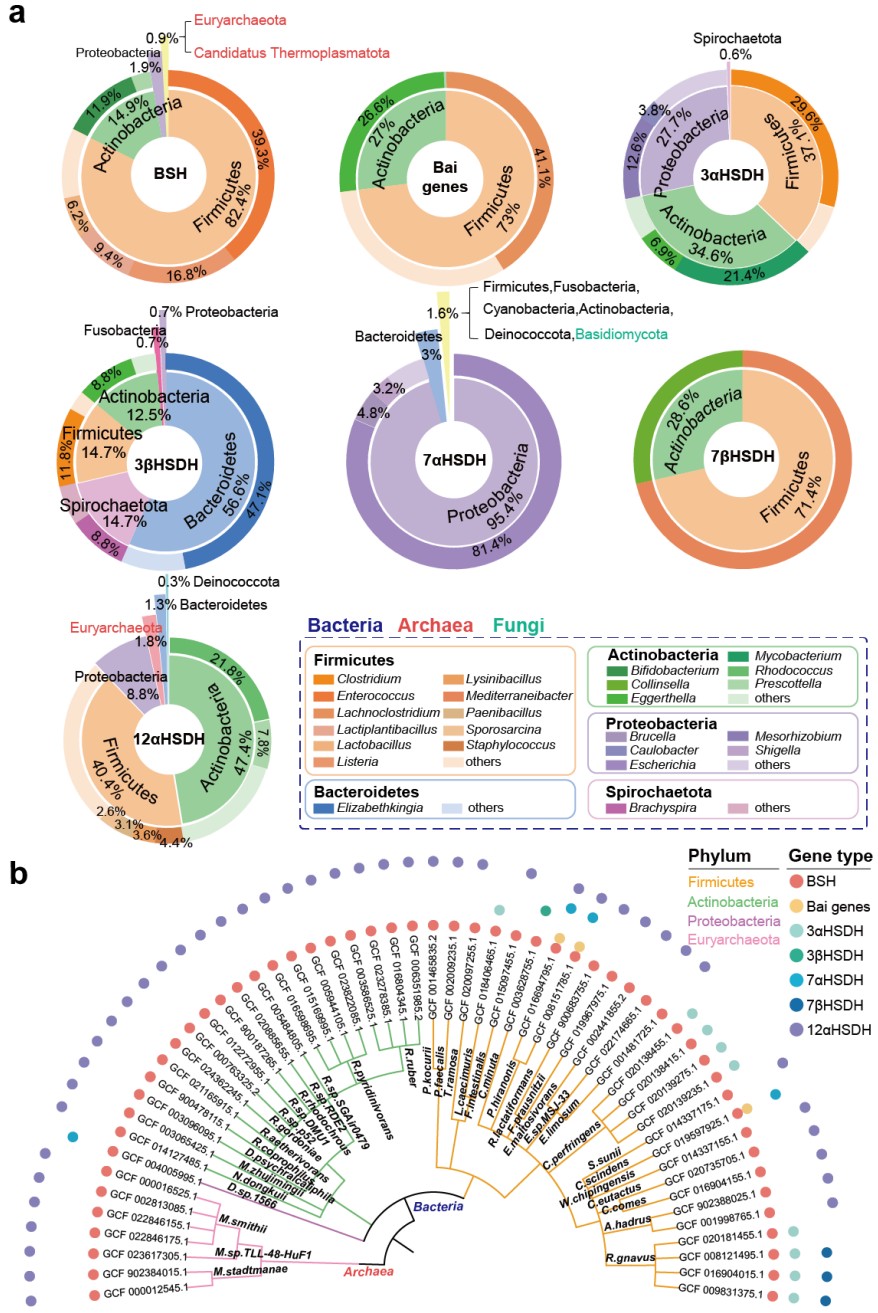

**FIG 2** Taxonomy distribution of secondary bile acid production genes. (a) Taxonomic composition of different secondary bile acid production genes. The pie charts show the proportions of genes across different phyla. The outer rings show the proportions of some main genera. (b) Phylogenetic tree of genomes carrying bile salt hydrolase and at least one bile acid-inducible genes/HSDHs. The branch colors represent different phyla. Symbols aligned to tree tips represent different types of secondary bile acid production genes.

Candidatus Thermoplasmatota (Table S2). Intrigued by the multi-kingdom distribution, we conducted a phylogenetic analysis, revealing similarities between bile salt hydrolase enzyme sequences in archaea Euryarchaeota and specific bacterial Firmicutes (Fig. S2a), suggesting a potential horizontal gene transfer (HGT) event.

Bile acid-inducible genes, which are involved in bile acid dehydroxylation, were found in only 22 species, predominantly from Firmicutes (176 genes, 73.0%) and Actinobacteria (65 genes, 27.0%) (Fig. 2a), aligning with previously reported sparse distribution (46). Due

to the cooperation nature of bile acid dehydroxylation involving multiple bile acid-inducible genes, we further explored the taxonomic distribution of genomes encompassing the full set of the bile acid-inducible genes. We identified 15 genomes from Firmicutes, primarily in *Lachnoclostridium* (*Clostridium scindens*: eight genomes, *Clostridium hylemonae*: three genomes) (Table S2).

The taxonomic distribution of HSDHs varied depending on their specific subtype (Fig. 2a). For example, 3αHSDH was predominantly present in *Clostridium* (47 genes, 29.6%) of Firmicutes, *Mycobacterium* (34 genes, 21.4%) of Actinobacteria, and *Mesorhizobium* (20 genes, 12.6%) of Proteobacteria. Within these groups, *Clostridium perfringens* and *Mycobacterium avium* were major carriers, with 88.5% and 97.1% of their genomes, respectively (Table S5). In contrast, 3βHSDH genes demonstrated a broader and distinctive distribution, predominantly within Bacteroidetes (77 genes, 56.6%) and Spirochaetota (20 genes, 14.7%). *Elizabethkingia anophelis* emerged as a significant carrier, with 3βHSDH genes found in all 50 genomes of this species (Table S5), indicating a core metabolic function. The 7αHSDHs displayed widespread distribution across seven bacterial phyla, with a staggering 95.4% (2,642 genes) in Proteobacteria. Notably, 97.9% (2,203 out of 2,250 genomes) of the genomes from *Escherichia coli* possessed this gene (Table S5). Interestingly, 7αHSDH was also found in the fungus *Rhizoctonia solani* of Basidiomycola, and phylogenetic analysis suggested its similarity with enzymes in Proteobacteria (Fig. S2b), indicating a potential origin of fungi 7αHSDH through HGT. In agreement with previous research (46), 7βHSDH enzymes were less prevalent. They were found primarily in *Ruminococcus gnavus* (four genes) and *Ruminococcus torques* (one gene) from Firmicutes, and *Collinsella aerofaciens* (two genes) from Actinobacteria. In contrast to other gene types that were commonly found in a high proportion of species, 7βHSDH genes were relatively strain specific, only presented in 16.7% (one out of six genomes) of *Collinsella aerofaciens* genomes and 75% (three out of four genomes) of *Ruminococcus gnavus* genomes (Table S5). In addition, 12αHSDH was predominantly found in Actinobacteria (183 genes, 47.4%) and Firmicutes (156 genes, 40.4%), mainly distributed in *Rhodococcus* (84 genes, 21.8%). A few genes were also detected in Euryarchaeota archaea, including *Methanobrevibacter smithii* (four genes), *Methanobrevibacter* sp. *TLL-48-HuF1* (one gene), and *Methanosphaera stadtmanae* (two genes). Phylogenetic analysis suggested that, similar to bile salt hydrolase, 12αHSDH genes in archaea may have been acquired from Firmicutes through HGT (Fig. S2c).

Our study uncovered 55 genomes from 37 species exhibited the simultaneous possession of bile salt hydrolase along with at least one bile acid-inducible genes or hydroxysteroid dehydrogenases. These genomes were considered to have a more independent capability for secondary bile acid production due to their rich repertoire of metabolic enzymes. These multi-functional genomes were primarily distributed among Firmicutes (28 genomes, 50.9%), Actinobacteria (19 genomes, 34.5%) in bacteria, and Euryarchaeota (seven genomes, 12.7%) in archaea. Notably, *Devosia* sp. *1566* was the only genome identified in Proteobacteria that exhibited multi-functional secondary bile acid metabolism potential. From the perspective of the diversity of secondary bile acid metabolism enzymes, microorganisms simultaneously possessing both bile salt hydrolase and 12αHSDH synthesis capabilities were common, with 43 genomes in 31 species displaying this trait. Moreover, genomes of Firmicutes showcased a greater diversity in secondary bile acid metabolism enzymes. Apart from the bile salt hydrolase–7αHSDH and bile salt hydrolase–12αHSDH combinations observed in other genomes, genomes from Firmicutes possessed five additional combinations involving other genes (Fig. 2b). These functionally rich genomes may play a significant role in the process of governing secondary bile acid production.

## Mammalian and environmental microbiota both served as reservoirs for secondary bile acid production genes

To achieve a comprehensive understanding of the distribution of genes involved in secondary bile acid metabolism across various hosts and environments, we conducted

a thorough analysis using the non-redundant Global Microbial Gene Catalogue (GMGC), a global-scale gene catalog constructed from worldwide metagenomes covering 14 habitats (47). This resource allowed us to examine the habitats and geographical locations of these genes, providing a global perspective on their distribution and comprehending their impact on One Health.

Our analysis revealed the widespread distribution of secondary bile acid production genes across a variety of global habitats. Notably, these genes were present not only in mammal hosts like humans, pigs, and mice, but also in diverse environmental settings such as wastewater, marine, and soil (Fig. 3a). A comparative analysis of the proportions of these secondary bile acid production genes in different habitats revealed intriguing variations in both prevalence and composition (Fig. 3b). Within mammal hosts, the prevalence of secondary bile acid production gene varied notably among organs, correlating with the primary sites of bile acid metabolism. The gut displayed the highest proportion of these genes compared to less metabolically active sites such as the oral and nasal cavities. Specifically, secondary bile acid metabolism genes account for 0.01126% (5,914 genes) of the human gut GMGC gene set, whereas the human oral cavity had only 0.00152% (203 genes). In environmental settings, the proportion of secondary bile acid production genes appeared to be influenced significantly by the human activities. Wastewater environments showed a higher proportion of these genes (404 genes, 0.01471%) compared to built environment (281 genes, 0.00346%), freshwater (10 genes, 0.00036%), soil (230 genes, 0.00030%), and marine settings (27 genes, 0.00003%). This suggested that microbial bile acid metabolism may be more prominent in environments with greater human impact. Moreover, we observed differences in the proportional composition of specific bile acid metabolism genes. For instance, bile salt hydrolase genes were more abundant in the guts of humans and pigs, whereas hydroxysteroid dehydrogenase genes were more prevalent in other environments.

This investigation highlighted that microbial bile acid metabolism was not restricted to specific organisms or ecosystems but rather a ubiquitous process across diverse hosts and environments. The variations in gene prevalence and composition likely reflected the adaptation of microbial communities to the available substrates or favored metabolic products in each habitat.

## Disease-specific alterations in the composition of secondary bile acid production genes

Considering the important role of bile acid metabolism in gastrointestinal and hepatic diseases, we performed a comprehensive analysis of secondary bile acid production profiles among inflammatory bowel disease (IBD), colorectal cancer (CRC), and NAFLD based on gut microbial genes and species.

We calculated a secondary bile acid metabolism differential score to quantify disruptions in secondary bile acid metabolism across diseases. This score incorporated the proportion of each metabolic gene type and the significance of their differences between disease and control groups. The analysis revealed significant disruptions of secondary bile acid metabolism, particularly in Crohn's disease (CD) and CRC, followed by ulcerative colitis (UC), while adenomas exhibited relatively smaller changes. Moreover, due to the gut-liver axis, NAFLD was also linked to changes in secondary bile acid metabolism (Fig. 4).

We further detailed the distinct differences in metabolic gene profiles across various diseases (Fig. 4). We observed distinct changes in the abundance of bile salt hydrolase genes, which serve as gateways for secondary bile acid metabolism. Bile salt hydrolase gene abundances were significantly downregulated in patients with CD and CRC. In particular, bile salt hydrolase-possessing microbes such as *Roseburia intestinalis*, *Anaerobutyricum hallii*, and *Blautia* sp. *SC05B48* consistently decreased (Table S8). In CD, *R. intestinalis* exhibited a higher relative abundance weighted by copy number (Fig. S3b), while *A. hallii* showed a higher relative abundance in CRC (Fig. S3e).

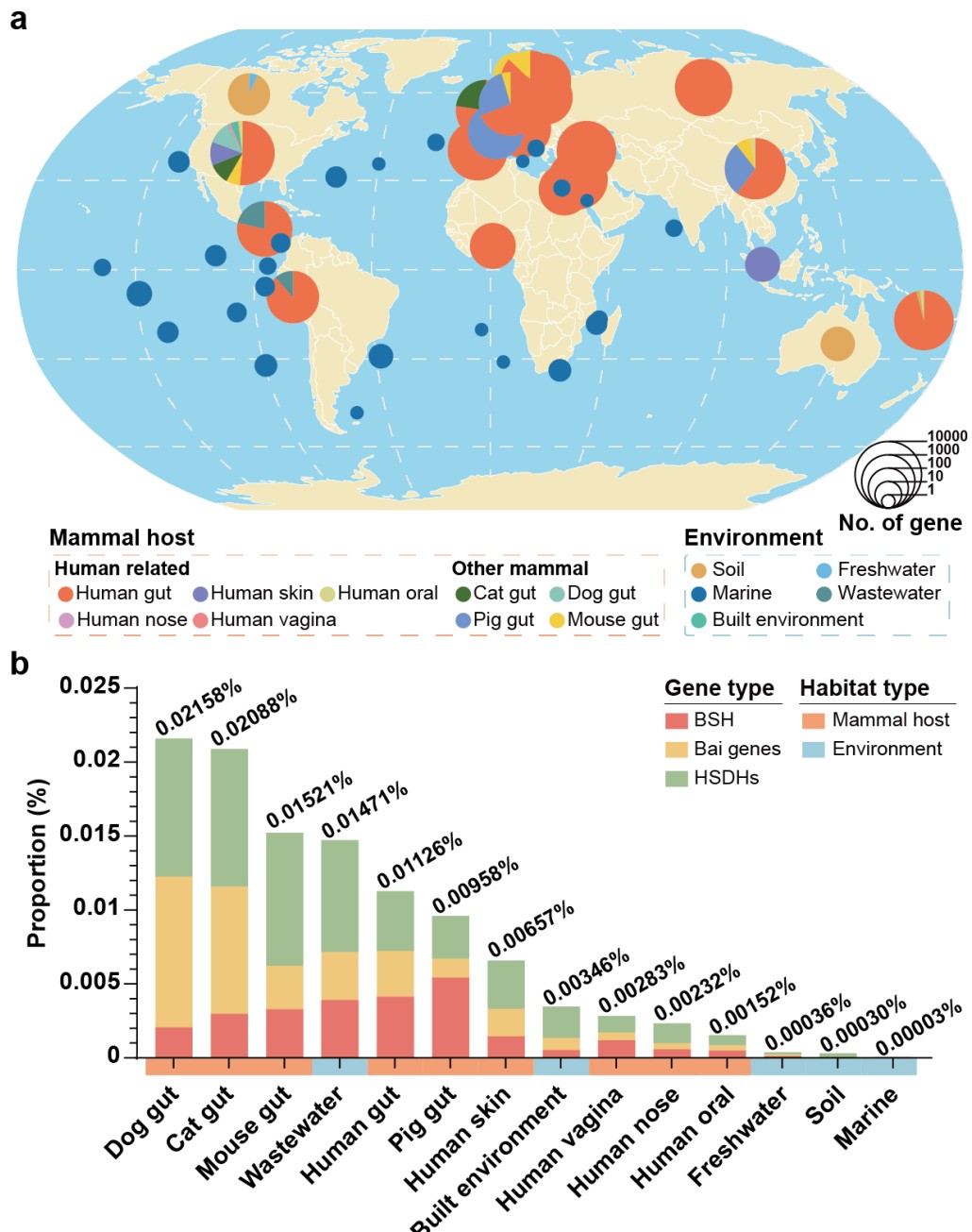

**FIG 3** Habitat distribution of secondary bile acid production genes. (a) Global map representing secondary bile acid production genes in GMGC. The size of the pie chart represents the number of secondary bile acid production genes. Different colors represent different habitats. The map was created in R using the "rnaturalearth" package (version 1.0.1) (48). (b) Secondary bile acid production genes composition in GMGC. Stacked bar chart shows proportions of bile salt hydrolase (red), bile acid-inducible genes (yellow), and HSDHs (green) in total genes of different habitats.

Next, we investigated the changes in genes involved in subsequent bile acid transformation, including dehydroxylation, oxidation, and epimerization. Significant disease-specific changes were observed. In detail, for bile acid-inducible genes crucial for producing DCA and lithocholic acid (LCA), their proportions were significantly increased in CD but decreased in NAFLD. This increase in CD was largely driven by *Ruminococcus gnavus* (Fig. S3c), while the decrease in NAFLD correlated with reduced *Eggerthella lenta* (Fig. S3g). α-HSDHs carry out the oxidation of the hydroxyl group at the 3-, 7-, and 12-carbons of cholic acid (CA) or chenodeoxycholic acid (CDCA). The proportion of 3αHSDH

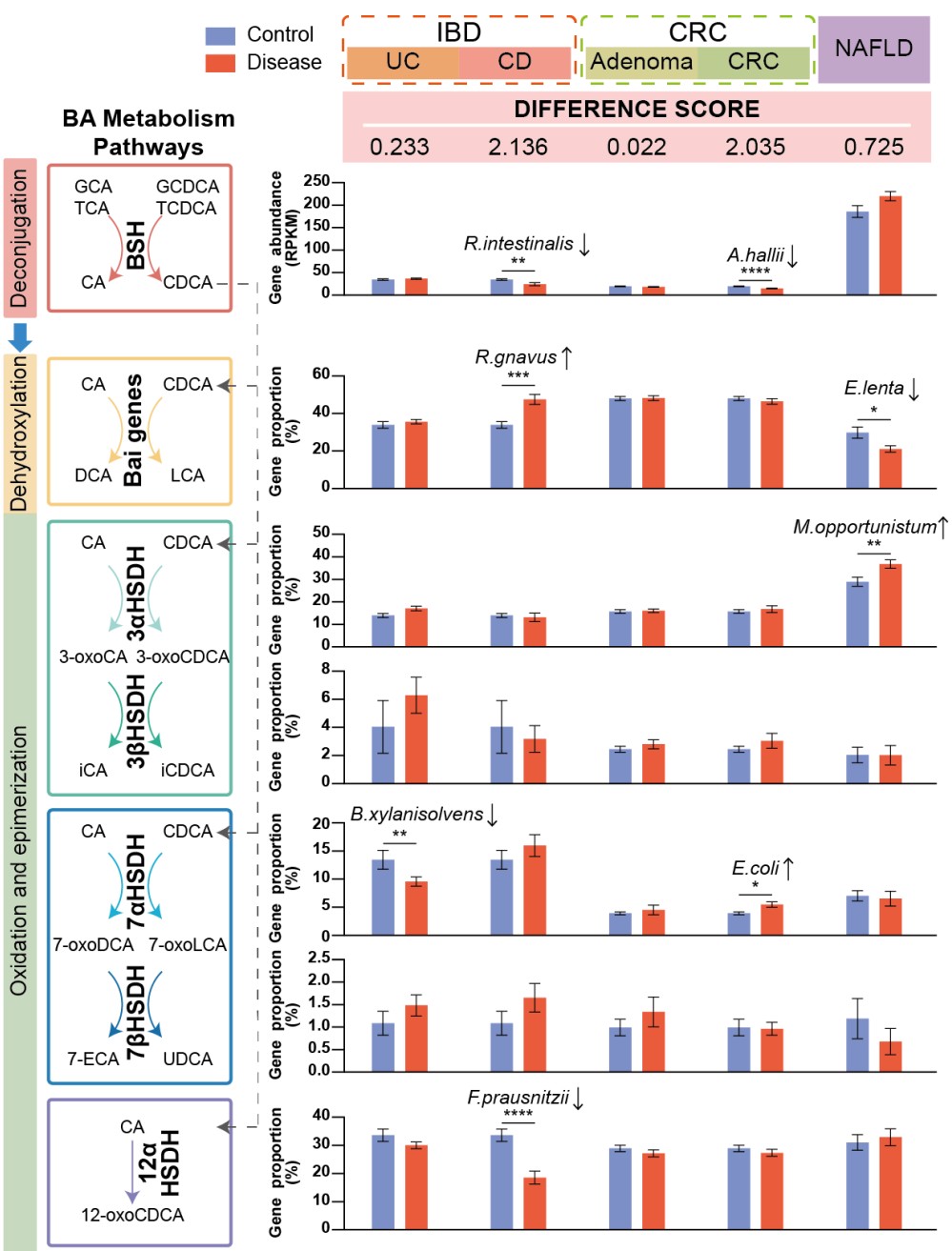

**FIG 4** Profiles of secondary bile acid production genes in intestinal and liver diseases. The bar plots show the gut microbial gene abundance of bile salt hydrolase as well as proportions of bile acid-inducible genes and HSDHs in different disease states. The microorganisms located above the bar plot are the major differential species with the highest weighted abundance among the gene-carrying microorganisms consistent with the gene alteration. The definition of "difference score" is in equation (3). Data are shown as mean with standard error (SE). The statistical differences between groups were determined by two-tailed Mann-Whitney U-test (UC, CD, adenoma, CRC) or paired $t$-test (NAFLD); the $P$ values were converted to asterisks (*$P$ < 0.05; **$P$ < 0.01; ***$P$ < 0.001, and ****$P$ < 0.0001). Numbers of samples by disease state for each study were IBD: control = 23, UC = 124, CD = 21; CRC: control = 63, adenoma = 47, CRC = 46; NAFLD: control = 10, NAFLD = 10.

genes significantly increased in NAFLD, corresponding to the rise of *Mesorhizobium opportunistum* (Fig. S3h). 7αHSDH genes exhibited disease-specific patterns. UC showed a decrease in the proportion of 7αHSDH, driven by *Bacteroides xylanisolvens* (Fig. S3a), while CRC displayed an increase, largely influenced by *Escherichia coli* (Fig. S3f). The 12αHSDH, primarily altered in CD, showed a decreased proportion linked to reduced

abundance of *Faecalibacterium prausnitzii* (Fig. S3d). Additionally, for 3βHSDH and 7βHSDH genes, which catalyze the reduction of bile acids after oxidation, did not exhibit significant changes in any of the diseases analyzed.

We further explored the species composition of secondary bile acid production genes across diseases using weighted relative abundance (Fig. S4). Firmicutes emerged as the dominant secondary bile acid producers in the gut, primarily in bile salt hydrolase, bile acid-inducible genes, and 12αHSDH. Furthermore, Actinobacteria were the main synthesizer of 3αHSDH, while Bacteroidetes predominantly contributed to 3βHSDH and 7αHSDH genes. Species like *Ruminococcus gnavus*, *Ruminococcus torques*, and *Collinsella aerofaciens*, which produce 7βHSDH, were also identified in the gut. Additionally, archaea producing bile salt hydrolase and 12αHSDH, as well as fungi producing 7αHSDH, identified in the gut also played roles in bile acid metabolism, contributing to the diversity and functionality of the gut microbiota.

## The metabolic process of secondary bile acids in pigs was more similar to that in humans

Animal models are critical in human medical research. To evaluate the feasibility of utilizing animal models for studying human secondary bile acid metabolism, we devised a scoring system to quantify the resemblance between the microbial genes involved in secondary bile acid production in various animals and humans. The similarity score for each animal model was derived by summing the weighted individual gene scores (a similarity score for individual gene) of each gene involved in the secondary bile acid metabolism.

Among the animal models evaluated, pigs displayed the highest overall similarity to humans in terms of secondary bile acid metabolism, followed by cats, while mice and dogs showed lower similarity scores (Fig. 5). This high resemblance in pigs was particularly evident in key enzymes, including bile salt hydrolase, 3αHSDH, 7βHSDH, and 12αHSDH, with 35.2% (764 genes), 31.7% (200 genes), 49.0% (24 genes), and 13.3% (132 genes) of the respective human genes being the same as those in pigs. In contrast, the number of "Unique" genes in humans was relatively small, with 146 (6.7%) bile salt hydrolase genes, 32 (5.1%) 3αHSDH genes, 1 (2.0%) 7βHSDH gene, and 64 (6.4%) 12αHSDH genes. Interestingly, while cats ranked second in the overall similarity, their microbiome demonstrated the highest similarity gene scores to humans for bile acid-inducible genes and 7αHSDH among these four animal models. Of the 1,627 bile acid-inducible genes and 304 7αHSDH genes in humans, 302 and 51 genes were identical to those found in cats, respectively. It was worth noting that the similarity of 3βHSDH between the animal models and humans was low. Only 56 genes were shared across the four animal models (41 genes in pigs, 33 genes in cats, 3 genes in mice, and 32 genes in dogs), and 32.6% (46 genes) of human intestinal 3βHSDH was unique.

These findings underscored the variable degree of similarity in secondary bile acid production genes between human and common animal models, highlighting the importance of selecting animal models that closely mimic human metabolic processes for other studies.

## DISCUSSION

Accurate and comprehensive gene annotation is essential for analyzing the functional and ecological roles of microbial communities, particularly in the context of One Health. In this study, we systematically identified genes involved in key stages of secondary bile acid metabolism, including deconjugation, dehydroxylation, oxidation, and epimerization, among 28,813 complete multi-kingdom microbial genomes sourced from the RefSeq database. Our expanded secondary bile acid production gene catalog builds upon and refines previous studies (8–16, 32, 34–41, 43, 49–53), addressing the limitations in available microbial and enzymatic resources. This enhancement allows for a more thorough exploration of the secondary bile acid metabolism from both taxonomic and

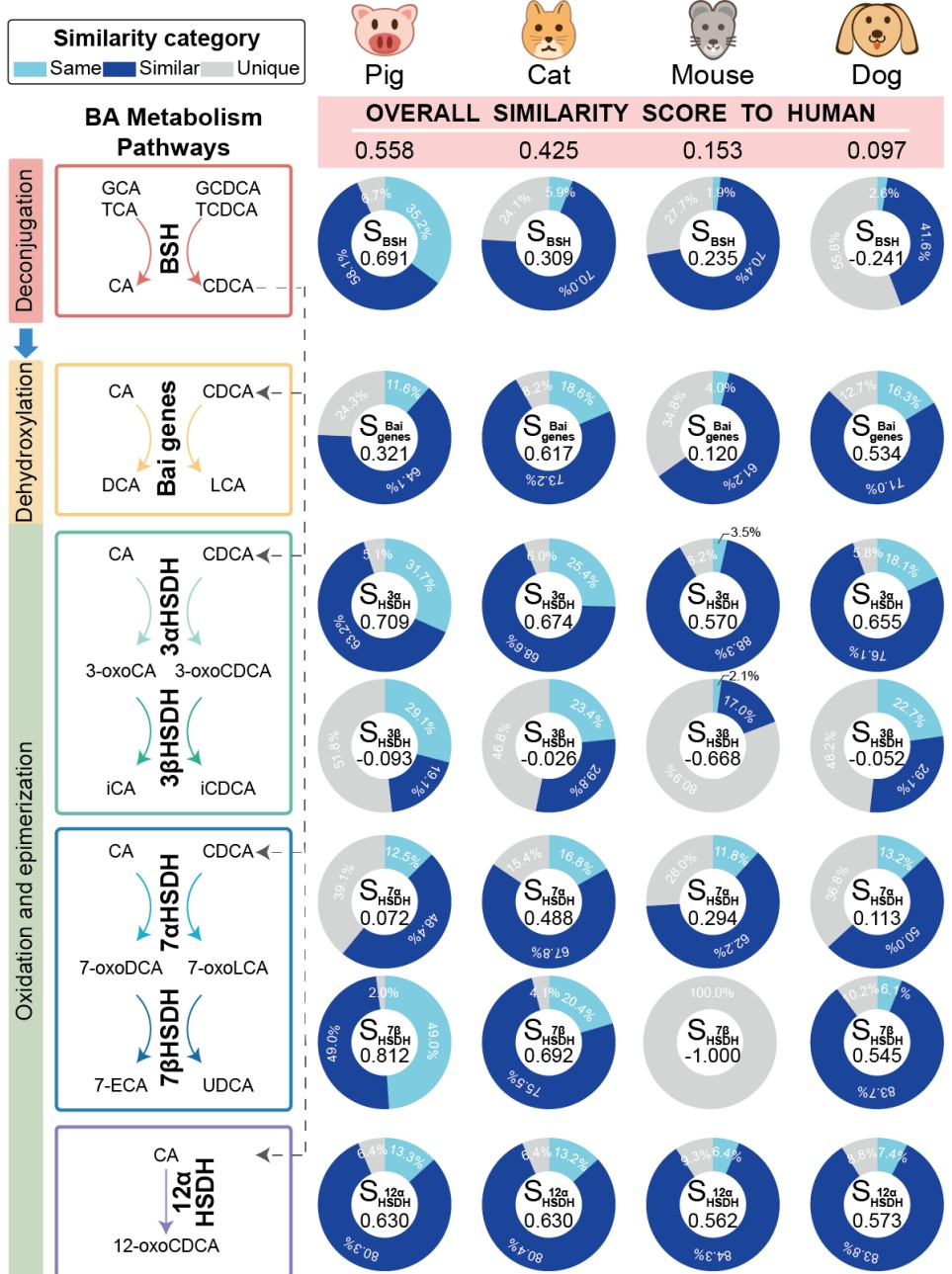

**FIG 5** Comparison of similarity of secondary bile acid production genes between humans and different animal models. The left panel illustrates the primary steps in the secondary bile acid metabolism pathway. Pie charts display the proportions of secondary bile acid production genes in three categories based on gene identity: light blue for the "Same" category, dark blue for the "Similar" category, and gray for the "Unique" category. The score in the center of each pie chart (e.g., $S_{BSH}$) represents gene score (equation [4]), which is a sum of weighted gene categories for each secondary bile acid gene. The panel on the right with a pink background shows the overall similarity scores for each corresponding animal model (equation [5]), a scoring system designed to assess the resemblance between animal models and humans in terms of secondary bile acid metabolism.

functional perspectives across diverse habitats, and significantly deepens our understanding of the ecological impact of these processes globally.

Our gene catalog revealed a broad taxonomic distribution of secondary bile acid production genes across multi-kingdom microorganisms, with a distinct tendency in the microbial participation at different stages of the metabolic pathway. This observation

suggests that effective secondary bile acid metabolism often requires inter-microbial cooperation. Notably, 55 genomes from 37 species demonstrated multifunctionality in secondary bile acid metabolism. Particularly, *Peptacetobacter hiranonis* from Firmicutes exhibited the most enriched functionality, possessing genes for bile salt hydrolase, bile acid-inducible genes, 7αHSDH, and 12αHSDH. This species, already identified as a biomarker for gastrointestinal functionality in dogs (54), showed strong and significant correlations with BaiCD, and the relative concentration of secondary bile acid in canine models (55). The multifunctional capacity of *P. hiranonis* and other key microorganisms highlights their potential as focal points for future research.

The microorganisms residing in diverse environments form highly interactive communities that facilitate interactions among humans, animals, and plants, playing indispensable roles in sustaining overall health and ecosystem stability (56). The variable frequency and composition of secondary bile acid production genes across different habitats underscored the critical interplay between microbial communities and human physiology. The widespread distribution of these genes in both mammalian hosts and human-impacted environments highlights how environmental factors shape micro-bial bile acid metabolism, with potential implications for disease pathogenesis. Mass spectrometry reanalyses (21) have detected microbe-derived bile acids in organs beyond the liver and intestines, including the skin (57), suggesting that bile acids may exert broader systemic roles. Comparative analyses of secondary bile acid production genes across different organs can facilitate a better understanding of the diverse physiological functions of bile acids. In addition to their well-known roles in digestion and metabo-lism, bile acids also function as signaling molecules, mediating communication between different organisms (58, 59). Thus, microbial bile acid metabolism in the environment can interfere with signaling systems by removing or transforming signaling compounds, having effects on the behaviors of both invertebrates and vertebrates (27). The variations in secondary bile acid production gene composition between different environmental settings may have ecological effects, highlighting the crucial role of microorganisms in One Health through intricate interactions with the ecosystem.

With the comprehensive understanding of the species and genes involved in bile acid metabolism, we could explore the related issues from a fresh perspective. One significant application is to systematically analyze the alterations in microbial bile acid metabolism under various pathological conditions. In our study, which included multiple cohorts from Europe, we focused on analyzing the profiles of secondary bile acid production genes to investigate their role in various diseases. Our findings uncovered distinct disease-specific changes in secondary bile acid metabolism. In patients with IBD, the capability of their gut microbiota to synthesize hydrophilic and less toxic secondary bile acids through α-HSDHs was impaired in both UC and CD. This reduction contributed to a more hydrophobic bile acid pool, which can disrupt intestinal epithelial integrity, leading to gut barrier dysfunction, a factor closely associated with IBD pathogenesis (60). Notably, CD exhibited a more pronounced overall change in secondary bile acid metabolism compared to UC, supporting theories of inflammation-associated and CD-specific metabolic disruptions (61). These insights offer new perspective on IBD pathogenesis and may guide more targeted research into the metabolic underpinnings of UC and CD. Furthermore, our study highlighted the importance of secondary bile acid metabolism in the progression from health to adenoma and ultimately CRC. While adenomas did not show significant metabolic shifts, CRC cases were characterized by a reduced capacity to synthesize secondary bile acids, accompanied by an increased proportion of 7αHSDH. The divergent trend of 7aHSDH compared to IBD may be related to the antagonistic effect of UDCA on farnesoid X receptor (FXR) (62). FXR inhibits CRC development by suppressing the Wnt/β-catenin pathway via the activation of TLE3 (63). Contrary to the reduced production of secondary bile acids in gastrointestinal diseases, NAFLD exhibited an increase in bile salt hydrolase genes, although this change was not statistically significant. Microbe-derived bile acids weakly activate FXR and notably reduced CDCA-induced FXR activation (64). Additionally, we noted a significant decrease

in the proportion of the bile acid-inducible genes responsible for generating DCA and LCA, both of which are potent FXR activators among microbe-derived bile acids. These findings align with our previous study highlighting the suppression of FXR-mediated signaling in NAFLD (20). The diminished FXR activity affects lipogenic genes and lipid absorption, contributing to the occurrence of NAFLD (65). Overall, alterations in secondary bile acid production genes likely impact disease development and progression by modifying the abundance, hydrophobicity, toxicity, and receptor interactions of secondary bile acids. This highlighted the potential of secondary bile acids and their metabolic pathways as targets for therapeutic intervention. Beyond assessing overall content of secondary bile acids, it is crucial to explore the compositional changes in different secondary bile acid types to fully understand their roles in diseases. Analyses based on fecal metagenomic sequencing provide insights into these changes but only offer a partial view. Therefore, the establishment of additional metabolomics cohorts is essential for a more comprehensive understanding of how disruptions in secondary bile acid metabolism affect different diseases.

Our gene catalog could also facilitate the comparison of secondary bile acid production genes across diverse habitats. This holds significant importance in selecting animal models suitable for studying bile acid metabolism in humans. Prior research has indicated the mirroring human intestinal physiology (66), dietary, and digestive patterns (67) of pigs, as well as the similar microbial composition in pig feces to humans (66), with 96% of the functional pathways from the human gut microbiome gene reference catalog appearing in the pig's catalog (68). It is worth noting that further validation is required regarding the availability of bile acid metabolism owing to the varying advantages for specific mechanisms among animal models. Our comparative analysis revealed that pigs, due to close resemblance of their microbial bile acid metabolism genes to those of humans, are particularly well suited for this purpose. Given these insights, future research should expand the comparative scope to include more animals like bears. It is essential to address the limitation of using the current GMGC by incorporating data from more species to capture the full complexity of microbial bile acid metabolism.

While our catalog represents a significant advancement in understanding secondary bile acid production genes, it does not yet capture the full diversity of microbial-derived bile acids recently reported (69). Emerging discoveries have expanded the known functions of bile acid metabolism enzymes, such as bile salt hydrolase, which has been shown to conjugate various amino acids to bile acids (70). Furthermore, the identification of novel enzymes, such as BAS-suc, responsible for producing 3-succinylated cholic acid (71), suggests that there is a vast, uncharted territory of microbial bile acid metabolism.

In conclusion, this systematic gene catalog derived from a vast array of microbial genomes substantially enriches the existing resource for microbial secondary bile acid production. This catalog not only enhances the identification of bile acid metabolism related genes in global microbiome but also provides a foundational resource for investigating the functional roles of microbial communities in various disease contexts. By applying our findings to disease studies and animal models, we offer fresh perspectives on how microbial bile acid metabolism influences health and disease. Ultimately, this work paves the way for innovative therapeutic approaches aimed at manipulating microbial metabolism to treat or prevent disease.

## MATERIALS AND METHODS

### Constructing the secondary bile acid production gene catalog

To better identify the target genes, HMM, an exhaustive algorithm based on dynamic programming (72), was used for its greater sensitivity compared to heuristic algorithms and capability to capture position-specific alignment information (73). HMM seed protein sequences of bile salt hydrolase, bile acid-inducible genes, and hydroxysteroid dehydrogenases derived from different species (Table S1) were obtained from the PubSEED

database (74, 75), and then aligned in Clustal Omega v1.2.4 (76) to construct HMMs on full-length proteins via hmmbuild (default mode, HMMER 3.3.2, hmmer.org), respectively. These model seed sequences were realigned to the models using hmmalign (default mode) before rebuilding models based on the alignments, and this iterative process was repeated three times to ensure the robustness of models. Subsequently, these HMMs were used to search for orthologs (hmmsearch [−tblout]) in the protein sequences of all 28,813 complete microbial genomes (bacteria: 28,324 genomes; archaea: 466 genomes; fungi: 23 genomes) provided by the National Center for Biotechnology Information (NCBI) RefSeq database (accessed in July 2022). Hits with an e-value less than $1e^{-65}$ were selected as candidates. To ensure high-quality sequences, further sorting was performed based on HMM score from hmmsearch results, and cutoffs (bile salt hydrolase: 400; BaiA: 390; BaiB: 1,000; BaiCD: 1,090; BaiE: 360; BaiF: 1,000; BaiG: 950; BaiH: 1,400; 3αHSDH: 300; 3βHSDH: 340; 7αHSDH: 325; 7βHSDH: 580; 12αHSDH: 250) at obvious HMM score drops were set to define secondary bile acid production genes. For bile acid-inducible genes and 7βHSDH, due to limited high-score results, genes from the PubSEED database were included as part of the secondary bile acid production gene catalog (Table S2). The DNA sequences corresponding to the secondary bile acid production gene catalog are in Table S3. Due to the redundancy in the protein sequences encoded by these genes, we constructed a non-redundant protein catalog based on their protein accession numbers (bile salt hydrolase: 416 sequences; bile acid-inducible genes: 146 sequences; 3αHSDH: 70 sequences; 3βHSDH: 62 sequences; 7αHSDH: 319 sequences; 7βHSDH: 3 sequences; 12αHSDH: 265 sequences, Table S4). This non-redundant protein catalog served as comprehensive and high-quality reference sequences for subsequent analysis of secondary bile acid production microbial enzymes in metagenomic data.

## Phylogenetic analyses of bile salt hydrolase, 7αHSDH, and 12αHSDH genes

To understand the evolution of bile salt hydrolase, 7αHSDH, and 12αHSDH distributed across different microbial kingdoms, multiple sequence alignments were performed by Clustal Omega v1.2.4 with non-redundant protein sequences of bile salt hydrolase, 7αHSDH, and 12αHSDH, and were trimmed with trimAl v1.4 (77) on automated1 mode. Next, maximum-likelihood phylogenetic trees were inferred by IQ-TREE v2.2.6 (78) using the suggested best-fit model (bile salt hydrolase: LG + R6; 7αHSDH: Q.pfam + R6; 12αHSDH: Q.pfam + R7) with 1,000 ultrafast bootstrap replicates and were visualized and annotated using iTOL (79).

## Analyzing the habitat distribution of secondary bile acid production gene catalog

For the distribution of secondary bile acid production genes at the global scale, DNA sequences of 302,655,267 non-redundant genes from 14 habitats in GMGC v1(47) were used as the query for BLASTX (80) searches against the non-redundant protein catalog of secondary bile acid production. Only blast hits with at least 70% identity and less than $1e^{-50}$ e-value were considered quality hits with bile acid metabolism capability (Table S6). By combining the habitat and geographic location information provided by the GMGC database for the sources of these genes, we analyzed the genes involved in secondary bile acid metabolism from both hosts such as human and pig, as well as environments like soil and ocean.

## Metagenomic data processing

Raw sequencing data were downloaded from the Sequence Read Archive (SRA) using the following identifiers: PRJEB1220 (IBD) for Nielsen et al. (81), PRJEB7774 (CRC) for Feng et al. (82), and PRJNA420817 (NAFLD) for Mardinoglu et al. (83) (in our study, paired "NAFLD" and "Control" samples were obtained from fecal samples of participants collected on day 0 and day 14, following an isocaloric low-carbohydrate diet with increased protein content for 14 days). Firstly, KneadData v.0.6 was utilized to remove

low-quality and contaminant reads including host-associated (hg38, felCat8, canFam3, mm10, rn6, susScr3, galGal4, and bosTau8 from UCSC Genome Browser) and laboratory-associated sequences by Trimmomatic v0.39 (84) (SLIDINGWINDOW:4:20 MINLEN:50 LEADING:3 TRAILING:3) and bowtie2 v.2.3.5 (85), respectively. Thereafter, filtered reads were used to generate taxonomic and functional profiles. Taxonomic classification of bacteria, archaea, fungi, and viruses was performed against our pre-built reference database using Kraken2 (86). The pre-built database comprises 32,875 bacterial, 489 archaeal, 11,694 viral reference genomes from the NCBI RefSeq database (accessed in August 2022), and 1,256 fungal reference genomes from the NCBI RefSeq database, FungiDB (http://fungidb.org) and Ensembl (http://fungi.ensembl.org) (accessed in August 2022). It was built using the Jellyfish program by counting distinct 31-mer in the reference libraries, with each k-mer in a read mapped to the lowest common ancestor of all reference genomes with exact k-mer matches. Furthermore, we used Bracken v.2.5.0 (87) to accurately count taxonomic abundance. The read counts of species were converted into relative abundance for further analysis. For function profiles, reads were assembled into contigs via Megahit v.1.2.9 (88) with "meta-sensitive" parameters, and Prodigal v.2.6.3 on the metagenome mode (−p meta) was then used to predict genes. Subsequently, we utilized CoverM V.4.0 to estimate gene abundance (-m rpkm) by mapping reads to the non-redundant reference constructed with CD-HIT using a sequence identity cutoff of 0.95 and a minimum coverage cutoff of 0.9 for the shorter sequences.

## Estimating the abundance of secondary bile acid production species and genes

Owing to the presence of multi-copy secondary bile acid production genes in certain genomes, we weighted the abundance of secondary bile acid production species by the average copy number (Table S7) of the genomes carrying secondary bile acid production gene within each species (see equations [1] and [2]) to mitigate the bias caused by the multiple copies. Additionally, we compared the weighted relative abundance of these secondary bile acid production microorganisms between diseases and control using two-tailed Mann-Whitney U-test (UC, CD, adenoma, CRC) or paired *t*-test (NAFLD) to find out the differential secondary bile acid production microorganisms (*P* value < 0.05). For differential secondary bile acid production microorganisms consistent with the alteration of corresponding genes between disease and control groups (Table S8), we defined the microorganisms with the highest weighted relative abundance as the major differential species.

$$\text{Average copy number} = \frac{\Sigma_{\text{Genome with SBA production gene of this species}} \text{ Copy number}}{\text{No. of genome with SBA production gene of this species}} \quad (1)$$

$$\text{Weighted relative abundance} = \text{Average copy number} * \text{Relative species abundance} \quad (2)$$

Based on BLASTX searches of genes in the non-redundant reference from metagenomic data against the non-redundant protein catalog of secondary bile acid production, we classified genes from metagenomes as secondary bile acid production genes if they exhibited at least 70% identity and an e-value of less than $1e^{-50}$. The abundance of a certain type of gene was estimated by the sum of the reads per kilobase per million mapped reads (RPKM) calculated by CoverM. Considering the entire metabolic process, we calculated the proportion of bile acid-inducible genes and hydroxysteroid dehydrogenases genes by considering their abundances as a whole. The abundance of bile salt hydrolase and proportion of bile acid-inducible genes or hydroxysteroid dehydrogenases between diseases and control were compared using two-tailed Mann-Whitney U-test (UC, CD, adenoma, CRC) or paired *t*-test (NAFLD) to calculate the *P* value for determining the significance. Subsequently, combining the importance of various genes and the significance level of their differences (*SL* in equation [3]), we defined a difference score

(see equation [3]) to assess the overall dysregulation of bile acid metabolism from a microbial perspective in diverse diseases.

$$\text{Difference score} = 0.5 * SL_{\text{BSH}} + 0.5 * \Sigma_{\text{Other SBA production genes}}(SL * \text{Proportion}) \qquad (3)$$

where $SL$ represents the significance level of the differences ($P \geq 0.05$ and rate of change > 10%: $SL = 0.5$; $P < 0.05$: $SL = 1$; $P < 0.01$: $SL = 2$; $P < 0.001$: SL = 3; $P < 0.0001$: $SL = 4$).

## Comparing secondary bile acid production genes in humans and other animal models

To assess the resemblance between animal models and humans in terms of microbial secondary bile acid metabolism, we developed a similarity score based on two primary factors: gene similarity and functional importance. This analysis was applied to genes involved in bile acid metabolism genes derived from human gut and various animal models, consistently identified within GMGC. First, we calculated the similarity scores for each gene involved in secondary bile acid metabolism by comparing the animal model to human (Gene score$_{\text{Animal model}}^{\text{Gene}}$, equation [4]). Then, these genes were weighted according to their significance in secondary bile acid metabolism, and the gene scores were summed to derive the overall similarity between the human and specific animal model (Overall score$_{\text{Animal model}}$, equation [5]).

To quantify the gene similarity, we categorized genes into three groups. Specifically, genes involved in secondary bile acid production were classified as "Same," "Similar," or "Unique" based on sequence identity comparison between the human and animal model gut microbiomes. In detail, DNA sequences of genes identified in GMGC as being involved in secondary bile acid metabolism were aligned using BLASTN to compare those from human gut microbiomes with those from the animal model gut microbiomes. Genes with the same gene ID in two systems (human versus specific animal model) were classified as "Same." For other genes, those showing at least 70% identity and an e-value of less than $1e^{-50}$ were classified as "Similar," while the remaining genes were classified as "Unique" to humans. In addition, to quantify how well animal models recapitulate the secondary bile acid metabolism pathways in humans, we calculated the ratios of human genes that were categorized as "Same," "Similar," and "Unique." Next, we assigned a specific score to each gene category, reflecting its degree of matching quality: a score of 1 for "Same," 0.7 for "Similar," and −1 for "Unique" (indicating a lack of close homologs in other animal models). Then, the gene score for each gene is calculated by summing the products of the ratios and their respective weights from each category (see equation [4]).

$$\text{Gene score}_i^j = \text{Ratio}(\text{Same}_{i,j}) + \text{Ratio}(\text{Similar}_{i,j}) * 0.7 - \text{Ratio}(\text{Unique}_{i,j}) \qquad (4)$$

where $i$ represents animal models, $i \in$ set (cat, dog, mouse, pig); $j$ represents secondary bile acid production genes, $j \in$ set (BSH, Bai genes, 3αHSDH, 3βHSDH, 7αHSDH, 7βHSDH, 12αHSDH).

Subsequently, we assessed the weights of each gene based on their contributions to the entire metabolic process. In detail, given that deconjugation of bile acids mediated by bile salt hydrolase is the gateway reaction to further modifications, bile salt hydrolase was assigned a weight of 0.5. The remaining four modification pathways collectively received a total weight of 0.5, with each pathway being allocated an equal weight of 0.125. Considering the collaboration between 3αHSDH and 3βHSDH, as well as 7αHSDH and 7βHSDH, these genes were assigned a weight of 0.0625 each, accounting for half of the weight of the respective pathway they involved.

Finally, to obtain a comprehensive assessment of the similarity between animal models and humans in terms of microbial secondary bile acid metabolism, we calculated

the overall score by integrating the individual gene scores, each weighted according to their assigned importance (see equation [5]).

$$\text{Overall score}_i = \Sigma_j W_j * \text{Gene score}_i^j \tag{5}$$

where $i$ represents animal models, $i \in$ set (cat, dog, mouse, pig); $j$ represents secondary bile acid production genes, $j \in$ set (BSH, Bai genes, 3αHSDH, 3βHSDH, 7αHSDH, 7βHSDH, 12αHSDH); $W$ is the weight of secondary bile acid production genes ($W_{BSH} = 0.5$, $W_{Bai\ genes} = 0.125$, $W_{3\alpha HSDH} = 0.0625$, $W_{3\beta HSDH} = 0.0625$, $W_{7\alpha HSDH} = 0.0625$, $W_{7\beta HSDH} = 0.0625$, $W_{12\alpha HSDH} = 0.125$).

## ACKNOWLEDGMENTS

This work was supported by the National Natural Science Foundation of China (grant number 92251307 to R.Z., 82170542 to R.Z., 82000536 to N.J.), the National Key Research and Development Program of China (grant number 2021YFF0703700/2021YFF0703702 to R.Z.), and the Guangdong Province "Pearl River Talent Plan" Innovation and Entrepreneurship Team Project (grant number 2019ZT08Y464 to L.Z.). The funders had no role in the study design, data collection and analysis, decision to publish, or preparation of the manuscript.

N.J., R.Z., and L.Z. conceived and designed the study. Y.Y. and W.G. drafted the manuscript. R.Z., L.T., W.C., X.Z. (Xinyue Zhu), M.S., T.X., T.Z., X.Z. (Xiaobai Zhang), L.Z., and N.J. reviewed and edited the manuscript. All authors read and approved the final manuscript.

## AUTHOR AFFILIATIONS

[1]Putuo People's Hospital, School of Life Sciences and Technology, Tongji University, Shanghai, China
[2]Shanghai Institute of Organic Chemistry, Chinese Academy of Sciences, Shanghai, China
[3]Research Institute, GloriousMed Clinical Laboratory Co, Ltd, Shanghai, China
[4]Department of General Surgery, The Six Affiliated Hospital, Guangdong Institute of Gastroenterology, Guangdong Provincial Key Laboratory of Colorectal and Pelvic Floor Diseases, Biomedical Innovation Center, Sun Yat-Sen University, Guangzhou, China
[5]State Key Laboratory of Genetic Engineering, Fudan Microbiome Center, School of Life Sciences, Fudan University, Shanghai, China

## AUTHOR ORCIDs

Yuwei Yang http://orcid.org/0009-0006-2258-2947
Ruixin Zhu http://orcid.org/0000-0002-5070-6453
Lixin Zhu http://orcid.org/0000-0001-7904-1769
Na Jiao http://orcid.org/0000-0003-3976-6313

## FUNDING

| Funder | Grant(s) | Author(s) |
| --- | --- | --- |
| National Natural Science Foundation of China | 92251307, 82170542 | Ruixin Zhu |
| National Key Research and Development Program of China | 2021YFF0703700/2021YFF0703702 | Ruixin Zhu |
| National Natural Science Foundation of China | 82000536 | Na Jiao |
| Guangdong Provincial Pearl River Talents Program | 2019ZT08Y464 | Lixin Zhu |

## DATA AVAILABILITY

The comprehensive secondary bile acid production gene catalog of bile salt hydrolases, bile acid-inducible genes, and hydroxysteroid dehydrogenases built based on RefSeq microbiome and its associated data including raw data, the taxonomic and habitat distribution, etc., are available within the paper and its supplemental material, as well as on GitHub (https://github.com/ywyang1/SBA-production-gene-catalog/). Genomes analyzed are available in the RefSeq database (https://ftp.ncbi.nlm.nih.gov/genomes/refseq/, accessed in July 2022). The sequence and metadata of the non-redundant genes from 14 habitats can be downloaded from GMGC v1.0 (https://gmgc.embl.de/download.cgi). The raw sequencing reads of metagenomic samples were deposited in SRA of the NCBI database under accession numbers PRJEB7774, PRJEB1220, and PRJNA420817. The scripts for secondary bile acid production gene catalog construction and further analyses are available on GitHub (https://github.com/ywyang1/SBA-production-gene-catalog/).

## ADDITIONAL FILES

The following material is available online.

### Supplemental Material

**Supplemental Figures (mSystems00817-24-s0001.docx).** Figures S1 to S4.
**Table S1 (mSystems00817-24-s0002.xlsx).** Seed sequence for HMM construction.
**Table S2 (mSystems00817-24-s0003.xlsx).** Secondary bile acid production gene catalog.
**Table S3 (mSystems00817-24-s0004.xlsx).** DNA sequences of the secondary bile acid production gene catalog.
**Table S4 (mSystems00817-24-s0005.xlsx).** Non-redundant protein catalog of secondary bile acid production.
**Table S5 (mSystems00817-24-s0006.xlsx).** Prevalence of secondary bile acid production genomes.
**Table S6 (mSystems00817-24-s0007.xlsx).** Genes with potential for secondary bile acid metabolism in GMGC.
**Table S7 (mSystems00817-24-s0008.xlsx).** Average copy number of secondary bile acid production species.
**Table S8 (mSystems00817-24-s0009.xlsx).** Differential secondary bile acid production microorganisms consistent with alteration of corresponding gene.

### Open Peer Review

**PEER REVIEW HISTORY (review-history.pdf).** An accounting of the reviewer comments and feedback.

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
