## [Reviewer comments · mSystems]

Systematic identification of secondary bile acid production genes in global microbiome

Yuwei Yang, Wenxing Gao, Ruixin Zhu, Liwen Tao, Wanning Chen, Xinyue Zhu, Mengping Shen, Tingjun Xu, Tingting Zhao, Xiaobai Zhang, Lixin Zhu, and Na Jiao

Corresponding Author(s): Na Jiao, Fudan University

Review Timeline:

Submission Date:	June 18, 2024
Editorial Decision:	September 1, 2024
Revision Received:	October 29, 2024
Accepted:	November 21, 2024

Editor: Naseer Sangwan

Reviewer(s): The reviewers have opted to remain anonymous.

Transaction Report:

DOI: <https://doi.org/10.1128/msystems.00817-24>

Re: mSystems00817-24 (Systematic identification of secondary bile acid production genes in global microbiome)

Dear Dr. Na Jiao:

Ensure you have read/considered the reviewer 2 comments attached as a separate document.

Revision Guidelines

- Upload point-by-point responses to the issues raised by the reviewers in a file named "Response to Reviewers," NOT in your cover letter.
- Upload a compare copy of the manuscript (without figures) as a "Marked-Up Manuscript" file.
- Upload a clean .DOC/.DOCX version of the revised manuscript and remove the previous version.
- Each figure must be uploaded as a separate, editable, high-resolution file (TIFF or EPS preferred), and any multipanel figures must be assembled into one file.

Minireviews are not subject to publication charges.

Author Bios: We encourage you to submit a biographical sketch of each author (limit of 150 words) along with a photo to be published at the end of your article. You can submit these with your modified manuscript.

Figures Enhancement: ASM has engaged a professional science illustrator, Patrick Lane of ScEYEnce Studios, to work with minireview authors at the modification stage to generate improved figures that are uniform throughout the journal. This art enhancement service is free of charge to authors of minireviews and full-length reviews, and turnaround time is fast. I think you will be pleased with the results. Please contact Patrick on receiving this letter. Complete contact information for Patrick and further instructions are posted at <https://journals.asm.org/pb-assets/pdf-text-excel-files/graphical-enhancement-support.pdf>.

Sincerely,
Naseer Sangwan
Editor
mSystems

Reviewer #1 (Comments for the Author):

The manuscript by Yang et al. describes a global analysis of bile acid metabolism across bacteria, archaea and fungal refseq genomes. They further look at the distribution of these species and genes across different environments (body site and geography). It is quite interesting that these genes are found outside of the gut, the primary ecosystem of study, in diverse settings such as soil, skin, nares and others.

Comments:

-Its hard to follow the acronyms throughout as there are so many. It would be easier for the reader if the secondary bile acids and some of the enzymes are written in full.

-Explain in the results text how you calculated the similarity score (figure 5)

-There is discussion of the distributions across different disease types but not linking specific SBAs to known biological outcomes or mechanisms of action. While pigs seem to be most similar, this is not surprising as the pig is known to be the closest animal model to humans outside non-human primates in terms of physiology and the microbiome.

-Further, while the gut is enriched for SBA genes, what could the role of these enzymes be in other environments?

-in the methods, did the authors control for number of genomes available per species? For example, certain species are over represented in databases which could bias the results. Please clarify in the methods.

The manuscript titled “Systematic identification of secondary bile acid production genes in global microbiome” reports research with much originality. However, the following comments should be addressed.

1. Line 96, “the advancements”, rather than “advancements”.
2. Line 97-99, hard to understand.
3. Line 100-102, “BaiE in whole-genome shotgun assembly sequences of human gut microbiomes, and $7\alpha/7\beta$ -HSDH in black bear fecal metagenomic datasets”, the sentences lacked of the predicate verb and were hard for understanding.
4. Line 104-105, “However, their study only considered 693 human gut microbial genomes”, the passive voice was recommended.
5. Line 161, “BA metabolism capabilities of various microbial species” means what?
6. Line 165, “At the genus taxonomy”.
7. Line 166, “distributed into genera”.
8. Line 184, “originating from phylum Firmicutes (176 genes, 73.0%) and Actinobacteria (65 genes, 27.0%)” should be modified as “originating from Firmicutes (176 genes, 73.0%) and Actinobacteria (65 genes, 27.0%)”; Line 188, “the Firmicutes”; Line 189, “the *Lachnoclostridium*”. Please revise throughout the manuscript.
9. Line 225, “the BSH and either the Bai genes or HSDHs, indicative of a more independent capability for SBA production”, misunderstanding description.
10. Figure 4. The microorganisms located above the bar plot are the major differential species possessing this gene. What criteria do you pick up for selecting the major one among so many species (Table S8 showed different genes located in various species).
11. Line 329-330, “which catalyze reduction after the oxidation by α -HSDH”. The sentence was hard for understanding.
12. Line 364-365, “However, cats’ microbiome showed higher similarity scores for Bai genes and 7α HSDH.” In the Figure 5, the similarity score for Bai genes was 0.617, the similarity score for 7α HSDH was 0.488, which were lower than 3α HSDH (0.674) and 7β HSDH (0.692). The description was not so accurate.
13. Line 402, “Human health is not isolated”, isolated means what?
14. Results parts: simply describe the findings in your figures and tables, and avoid excess and inaccurate descriptions.
15. Finally, there are many language mistakes. Thus, the language should be polished by native speakers.

Reviewers' Comments:

Reviewer #1:

Comments for the Author:

The manuscript by Yang et al. describes a global analysis of bile acid metabolism across bacteria, archaea and fungal refseq genomes. They further look at the distribution of these species and genes across different environments (body site and geography). It is quite interesting that these genes are found outside of the gut, the primary ecosystem of study, in diverse settings such as soil, skin, nares and others.

Answer: We sincerely appreciate the reviewer's insightful and positive feedback.

Q1: Its hard to follow the acronyms throughout as there are so many. It would be easier for the reader if the secondary bile acids and some of the enzymes are written in full.

Answer: We thank the reviewer for the constructive comment. Revisions have been made throughout the manuscript in response to this suggestion.

Q2: Explain in the results text how you calculated the similarity score (figure 5)

Answer: We apologize for not providing the detailed calculation process for the similarity score presented in Figure 5.

The overall similarity in secondary bile acid metabolism between the animal models and humans, as defined in this study, involves two primary factors: gene similarity and functional importance. First, we calculated the similarity scores for each gene involved in secondary bile acid metabolism by comparing the animal model to that of human (denoted as $\text{Gene-score}_{\text{Animal model}}^{\text{Gene}}$, equation (1)). Then, each gene was assigned a weight according to its functional relevance in secondary bile acid metabolism and these weighted Gene-scores were summed to obtain the overall similarity score between the human and specific animal model (denoted as $\text{Overall-score}_{\text{Animal model}}$, equation (2)).

To further clarify, these steps are as follows:

Step One:

The similarity scores for each gene ($\text{Gene-score}_{\text{Animal model}}^{\text{Gene}}$) was defined mainly according to the identity of genes involved in secondary bile acid metabolism between humans and animal models. To quantify the gene similarity, we categorized genes into three groups (Fig. R1). Specifically, genes involved in secondary bile acid production were classified as ‘Same’, ‘Similar’ or ‘Unique’ based on sequence identity comparison between the human and animal model gut microbiomes. In detail, DNA sequences of genes identified in GMGC as being involved in secondary bile acid metabolism were aligned using BLASTN to compare those from human gut microbiomes with those from the animal model gut microbiomes. Genes with the same gene ID in two systems (human versus specific animal model) were classified as ‘Same’. For other genes, those showing at least 70% identity and an e-value of less than $1e^{-50}$ were classified as ‘Similar’, while remaining genes were classified as ‘Unique’ to humans. In addition, to quantify how well animal models recapitulate the secondary bile acid metabolism pathways in humans, we calculated the ratios of human genes that were categorized as ‘Same’, ‘Similar’ and ‘Unique’. Next, we assigned specific score to each gene category, reflecting its degree of matching quality: a score of 1 for ‘Same’, 0.7 for ‘Similar’, and -1 for ‘Unique’ (indicating a lack of close homologs in other animal models). Then, the Gene-score for each gene is calculated by summing the products of the ratios and their respective weights from each category using the following equation:

$$\text{Gene-score}_i^j = \text{Ratio}(\text{Same}_{i,j}) + \text{Ratio}(\text{Similar}_{i,j}) * 0.7 - \text{Ratio}(\text{Unique}_{i,j}) \quad (1)$$

where i represents animal models, $i \in \text{set}(\text{cat, dog, mouse, pig})$; j represents secondary bile acid production genes, $j \in \text{set}(\text{BSH, Bai genes, } 3\alpha\text{HSDH, } 3\beta\text{HSDH, } 7\alpha\text{HSDH, } 7\beta\text{HSDH, } 12\alpha\text{HSDH})$

FIG R1. The scheme for defining Same, Similar, and Unique categories based on secondary bile acid gene sequence identity between the animal models and humans. The secondary bile acid production genes were categorized into three distinct groups at different similarity level based on the sequence comparison of secondary bile acid production genes between humans and animal models. Each similarity category was assigned with a score to reflect the match quality.

Step Two:

We assessed weights of each gene based on their contributions to the entire secondary bile acid metabolic process (Fig. R2). In detail, given that deconjugation of bile acids mediated by bile salt hydrolase is the gateway reaction to further modifications, bile salt hydrolase was assigned a weight of 0.5. The remaining four modification pathways collectively received a total weight of 0.5, with each pathway being allocated an equal weight of 0.125. Considering the collaboration between 3α HSDH and 3β HSDH, as well as 7α HSDH and 7β HSDH, these genes were assigned weights of 0.0625, accounting for half of the weight of the respective pathway they involved. The weight of each gene was distributed as follow:

$$W_{\text{BSH}} = 0.5, W_{\text{Bai genes}} = 0.125, W_{3\alpha\text{HSDH}} = 0.0625, W_{3\beta\text{HSDH}} = 0.0625,$$

$$W_{7\alpha\text{HSDH}} = 0.0625, W_{7\beta\text{HSDH}} = 0.0625, W_{12\alpha\text{HSDH}} = 0.125$$

FIG R2. Major secondary bile acid metabolism pathways including deconjugation, dehydroxylation, and epimerization. The deconjugation of conjugated primary bile acids is mediated by bile salt hydrolase (BSH) to produce unconjugated primary bile acids. Subsequently, these primary bile acids are converted to secondary bile acids through dehydroxylation mediated by bile acid-inducible (Bai) genes or epimerization mediated by α/β -hydroxysteroid dehydrogenases (α/β -HSDHs).

Finally, the overall similarity scores of each animal model (Overall-score_{Animal model}) for comprehensive assessment of the similarity to humans in terms of microbial secondary bile acid metabolism was calculated by integrating the individual Gene-scores, each weighted according to their assigned importance:

$$Overall-score_i = \sum_j W_j * Gene-score_i^j \quad (2)$$

where i represents animal models, $i \in \text{set}(\text{cat}, \text{dog}, \text{mouse}, \text{pig})$; j represents secondary bile acid production genes, $j \in \text{set}(\text{BSH}, \text{Bai genes}, 3\alpha\text{HSDH}, 3\beta\text{HSDH}, 7\alpha\text{HSDH}, 7\beta\text{HSDH}, 12\alpha\text{HSDH})$;

Taking the comparison of secondary bile acid production genes between humans and pigs as an example, 764 human bile salt hydrolase genes are the ‘Same’, 1259 are ‘Similar’, and 146 are ‘Unique’. Therefore, the single similarity score of pig in bile salt hydrolase is 0.691 ($Gene-score_{Pig}^{BSH} = 764/2169 + 1259/2169*0.7 - 146/2169 =$

0.691). Similarly, scores for other genes can be calculated (e.g. $Gene\text{-}score_{Pig}^{Bai\ genes} = 0.321$, $Gene\text{-}score_{Pig}^{3\alpha HSDH} = 0.709$, $Gene\text{-}score_{Pig}^{3\beta HSDH} = -0.093$, $Gene\text{-}score_{Pig}^{7\alpha HSDH} = 0.072$, $Gene\text{-}score_{Pig}^{7\beta HSDH} = 0.812$, $Gene\text{-}score_{Pig}^{12\alpha HSDH} = 0.630$). According to the defined weights, the overall similarity score of pigs is 0.558 ($Overall\text{-}score_{Pig} = 0.5*0.691 + 0.125*0.321 + 0.0625*0.709 + 0.0625*(-0.093) + 0.0625*0.072 + 0.0625*0.812 + 0.125*0.630 = 0.558$).

We have included a detailed explanation of the similarity score calculation in the Methods section (Lines 611-658). Additionally, the descriptions in the Results (Lines 341-347) and the legend of Figure 5 have been modified to enhance clarity.

Q3: There is discussion of the distributions across different disease types but not linking specific SBAs to known biological outcomes or mechanisms of action. While pigs seem to be most similar, this is not surprising as the pig is known to be the closest animal model to humans outside non-human primates in terms of physiology and the microbiome.

Answer: We appreciate the reviewer's critical comments.

We have elaborated in the revised manuscript to link specific secondary bile acids to known biological outcomes or mechanisms of action (Lines 423-447). Changes in the hydrophobicity, toxicity, and receptor interactions of the secondary bile acid pool can affect the occurrence of diseases such as IBD, NAFLD, and CRC. Specifically, the increase in the hydrophobicity of bile acids leads to damage to intestinal epithelial cells, resulting in dysfunction of the intestinal barrier and may facilitate the development of IBD¹. CRC might be linked to the alteration in the Wnt/ β -catenin pathway, potentially initiated by the increase in the ratio of UDCA leading to FXR inhibition². The onset of NAFLD may be related to disruptions in lipid metabolism caused by reduced FXR-activate bile acid levels³.

Pigs are considered excellent animal models of human health and disease due to their similarities in terms of physiology and the microbiome. However, this does not mean that pigs necessarily mirror human mechanisms in all aspects. For example, rats

demonstrate a closer resemblance to humans in regulation of placentation and trophoblast invasion than pigs⁴. In this study, we aimed to assess the feasibility of animal models including pigs, cats, mice and dogs in microbial bile acid metabolism. We found that pigs exhibited the highest overall similarity to humans in this regard by quantifying the resemblance between the intestinal microbial secondary bile acid production genes of various animals and humans. It is worth noting that cats displayed the most similar bile acid-inducible genes and 7 α HSDH to humans among these four animal models considered. This indicates that a comprehensive utilization of different animal models may be essential in the context of practical applications. And we also revised our description in the Discussion section (Lines 458-468).

Q4: Further, while the gut is enriched for SBA genes, what could the role of these enzymes be in other environments?

Answer: We thank the reviewer for the insightful comment. The One Health concept emphasizes the interconnectedness of human health and well-being with the health of other ecosystem components, including soil, plants, and animals⁵. Therefore, our investigation delved into the bile acid metabolism of microorganisms across multiple habitats to elucidate its pivotal role in the broader ecosystem. Bile acids, especially secondary bile acids derived from the microbial metabolism in the mammal gut, has been widely studied due to their important role in maintaining the health of hosts through affecting metabolism, immunity, and other mechanisms⁶. Therefore, we have not only investigated human intestinal but also studied species like pigs and mice to understand this metabolism in their intestines, and evaluated their similarity to the human in terms of bile acid metabolism. Furthermore, bile acids can reflux into the environment such as soil and water along with human or animal feces and urine⁷, serving as carbon- and energy-rich growth substrates for environmental microorganisms⁸. The bile acid metabolism occurred in the environment not just regulates the microbiota and environment directly⁹, but also has an impact on other organisms. On one hand, bile acids can act as signaling compounds connecting various organisms^{10,11}. As such, microbial bile acid metabolism in the environment

could interfere with signaling systems by removing or transforming signaling compounds. On the other hand, these metabolic products may circulate among different components of the ecosystem through food webs to influence the cycle and diversity of microbial-derived, exerting their effects on a larger scale. To sum up, these enzymes in the environment play crucial ecological roles, further impacting One Health. Relevant descriptions have been added in the Introduction and Discussion part (Lines 81-93, 407-415).

Q5: in the methods, did the authors control for number of genomes available per species? For example, certain species are over represented in databases which could bias the results. Please clarify in the methods.

Answer: Our decision to incorporate all complete genomes of bacteria, archaea, and fungi retrieved from RefSeq in July 2022 into this study was driven by the variations observed in secondary bile acid metabolism genes among certain species across their genomes (Supplementary Table 2). This inclusion not only guarantees the comprehensiveness and sequence diversity of the gene catalog we constructed, but also allows for better understanding of genome-specific gene-diversity and copy numbers.

We totally agree with the concern you raised and have given careful consideration to the potential bias in our results. Appropriate measures have been implemented to address any bias caused by imbalance of genomes in certain species in our findings. In Supplementary Table 5, we provided the proportion of genomes carrying secondary bile acid-production gene in the total genomes of the corresponding species to compare the gene prevalence. Moreover, we weighted the species abundance in fecal metagenomic sequencing samples based on the average copy number of the genomes carrying secondary bile acid production gene within each species (Lines 576-580) to fairly evaluate the contributions of different species to bile acid metabolism (Supplementary Fig. 4).

Reviewer #2:

Comments for the Author:

The manuscript titled “Systematic identification of secondary bile acid production genes in global microbiome” reports research with much originality. However, the following comments should be addressed.

Answer: We are grateful for the reviewer's positive comments and have taken steps to address all raised concerns.

Q1: *Line 96, “the advancements”, rather than “advancements”.*

Answer: We thank the reviewer’s detailed suggestion for language refinement and have implemented the correction accordingly (Line 100). Additionally, we have thoroughly reviewed the manuscript to ensure similar issues are avoided.

Q2: *Line 97-99, hard to understand.*

Answer: We apologize for the poor sentence organization. Our aim was to summarize existing research on the identification of secondary bile acid metabolism genes using bioinformatics methods. We have revised the sentences to enhance readability (Lines 100-104).

Q3: *Line 100-102, “BaiE in whole-genome shotgun assembly sequences of human gut microbiomes, and 7 α /7 β -HSDH in black bear fecal metagenomic datasets”, the sentences lacked of the predicate verb and were hard for understanding.*

Answer: The confusion primarily arose from lack of the predicate verb and long sentences. We have re-written the relevant sections to improve clarity and readability (Lines 105-108).

Q4: *Line 104-105, “However, their study only considered 693 human gut microbial genomes”, the passive voice was recommended.*

Answer: We appreciate the reviewer’s language polishing recommendation. The corresponding modification has been made in the revised manuscript (Line 110).

Q5: Line 161, “BA metabolism capabilities of various microbial species” means what?

Answer: Sorry for the previous ambiguous description. We aimed to compare the types of secondary bile acid metabolism enzymes in different species and even different strains to reflect their varying metabolism capabilities. Relevant content has been modified in the revised manuscript (Lines 169-171).

Q6: Line 165, “At the genus taxonomy”.

Answer: We sincerely thank the reviewer for this helpful modification and revised into the more common expression “At the genus level” accordingly (Line 176).

Q7: Line 166, “distributed into genera”.

Answer: We appreciate the reviewer’s professional language revision. We have carefully incorporated the suggested modifications into our manuscript (Line 177).

Q8: Line 184, “originating from phylum Firmicutes (176 genes, 73.0%) and Actinobacteria (65 genes, 27.0%)” should be modified as “originating from Firmicutes (176 genes, 73.0%) and Actinobacteria (65 genes, 27.0%)”; Line 188, “the Firmicutes”; Line 189, “the Lachnoclostridium”. Please revise throughout the manuscript.

Answer: Thank you for pointing this out. We have carefully addressed this issue throughout the paper.

Q9: Line 225, “the BSH and either the Bai genes or HSDHs, indicative of a more independent capability for SBA production”, misunderstanding description.

Answer: We apologize for the inaccurate description. The microbial transformations of bile acids primarily involve deconjugation by bile salt hydrolases, dehydroxylation by proteins encoded by bile acid-inducible genes, oxidation and epimerization by α/β -HSDHs, with deconjugation being the gateway reaction¹². We hypothesized that genomes capable of both deconjugation and at least one of dehydroxylation, oxidation,

or epimerization would exhibit a more comprehensive and independent capacity for secondary bile acid production. Therefore, we further explored the distribution of genomes carrying bile salt hydrolase along with at least one bile acid-inducible gene or HSDHs. These clarifications and modifications have been incorporated into the revised manuscript (Lines 232-236).

Q10: Figure 4. The microorganisms located above the bar plot are the major differential species possessing this gene. What criteria do you pick up for selecting the major one among so many species (Table S8 showed different genes located in various species).

Answer: We apologize for the lack of clarity regarding the definition of the major differential species. For each metabolism gene, we calculated the weighted relative abundance of species possessing this gene based on the average gene copy number, and identified the differential secondary bile acid-production microorganisms consistent with the significant differences of corresponding gene (Table S8). The species with the highest weighted relative abundance was defined as the major differential species. Details of the selecting criteria have been incorporated in the revised manuscript (Lines 584-587), and the figure legend was modified accordingly.

Q11: Line 329-330, “which catalyze reduction after the oxidation by α -HSDH”. The sentence was hard for understanding.

Answer: Sorry for the poorly organized sentence. Our intention was to explain that β -HSDHs catalyze the reduction of bile acids following their oxidation by the corresponding α -HSDH. We have modified the sentence accordingly to reflect this clarification (Lines 324-326).

Q12: Line 364-365, “However, cats’ microbiome showed higher similarity scores for Bai genes and 7 α HSDH.” In the Figure 5, the similarity score for Bai genes was 0.617, the similarity score for 7 α HSDH was 0.488, which were lower than 3 α HSDH (0.674) and 7 β HSDH (0.692). The description was not so accurate.

Answer: We apologize for any confusion caused by the unclear description.

Our intention was to explain that, in comparing the similarity of various microbial bile acid metabolism enzymes across the four animal models to those in humans, we calculated an overall similarity score for each animal model. This overall score represented the sum of the similarity scores of all genes involved in bile acid metabolism. Pigs exhibited the highest overall similarity to humans, with an overall score of 0.558, while cats displayed a score of 0.425.

In addition, we compared single similarity scores for each bile acid metabolism enzyme. From this perspective, we discovered that the bile acid-inducible genes and 7 α HSDH of cats' microbiome exhibited the higher similarity scores than others. These points have been addressed and clarified in the revised manuscript (Lines 356-358).

Q13: Line 402, "Human health is not isolated", isolated means what?

Answer: We intended to convey the concept of One Health emphasizing the inseparable link between human and other components of the ecosystem, as well as the contribution of microbiome to this overall well-being¹³. Building upon this foundation, we further elaborated on the specific impact of bile acid metabolism mechanisms. Relevant modification has been included in the revised manuscript (Lines 395-397).

Q14: Results parts: simply describe the findings in your figures and tables, and avoid excess and inaccurate descriptions.

Answer: Thank you for this crucial comment. As suggested by the reviewer, we have simplified the Results part and corrected the inappropriate descriptions.

Q15: Finally, there are many language mistakes. Thus, the language should be polished by native speakers.

Answer: We have carefully addressed the issues related to writing and grammar throughout the paper with the help of native speakers.

References:

- 1 Stenman, L. K., Holma, R., Eggert, A. & Korpela, R. A novel mechanism for gut barrier dysfunction by dietary fat: epithelial disruption by hydrophobic bile acids. *Am J Physiol Gastrointest Liver Physiol* **304**, G227-234 (2013). <https://doi.org/10.1152/ajpgi.00267.2012>
- 2 Dong, X., Cai, C. & Fu, T. FXR suppresses colorectal cancer by inhibiting the Wnt/beta-catenin pathway via activation of TLE3. *Genes Dis* **10**, 719-722 (2023). <https://doi.org/10.1016/j.gendis.2022.09.006>
- 3 Clifford, B. L. *et al.* FXR activation protects against NAFLD via bile-acid-dependent reductions in lipid absorption. *Cell Metab* **33**, 1671-1684 e1674 (2021). <https://doi.org/10.1016/j.cmet.2021.06.012>
- 4 Zang, X. *et al.* Cross-Species Insights into Trophoblast Invasion During Placentation Governed by Immune-Featured Trophoblast Cells. *Adv Sci (Weinh)*, e2407221 (2024). <https://doi.org/10.1002/advs.202407221>
- 5 Banerjee, S. & van der Heijden, M. G. A. Soil microbiomes and one health. *Nat Rev Microbiol* **21**, 6-20 (2023). <https://doi.org/10.1038/s41579-022-00779-w>
- 6 Cai, J., Sun, L. & Gonzalez, F. J. Gut microbiota-derived bile acids in intestinal immunity, inflammation, and tumorigenesis. *Cell Host Microbe* **30**, 289-300 (2022). <https://doi.org/10.1016/j.chom.2022.02.004>
- 7 Ridlon, J. M., Kang, D. J. & Hylemon, P. B. Bile salt biotransformations by human intestinal bacteria. *J Lipid Res* **47**, 241-259 (2006). <https://doi.org/10.1194/jlr.R500013-JLR200>
- 8 Holert, J. *et al.* Metagenomes Reveal Global Distribution of Bacterial Steroid Catabolism in Natural, Engineered, and Host Environments. *mBio* **9** (2018). <https://doi.org/10.1128/mBio.02345-17>
- 9 Mansell, D. S. *et al.* Fate of endogenous steroid hormones in steer feedlots under simulated rainfall-induced runoff. *Environ Sci Technol* **45**, 8811-8818 (2011). <https://doi.org/10.1021/es202072f>
- 10 Buchinger, T. J., Li, W. & Johnson, N. S. Bile salts as semiochemicals in fish. *Chem Senses* **39**, 647-654 (2014). <https://doi.org/10.1093/chemse/bju039>
- 11 Doyle, W. I. *et al.* Faecal bile acids are natural ligands of the mouse accessory olfactory system. *Nat Commun* **7**, 11936 (2016). <https://doi.org/10.1038/ncomms11936>
- 12 Guzior, D. V. & Quinn, R. A. Review: microbial transformations of human bile acids. *Microbiome* **9**, 140 (2021). <https://doi.org/10.1186/s40168-021-01101-1>
- 13 Ma, L. C., Zhao, H. Q., Wu, L. B., Cheng, Z. L. & Liu, C. Impact of the microbiome on human, animal, and environmental health from a One Health perspective. *Sci One Health* **2**, 100037 (2023). <https://doi.org/10.1016/j.soh.2023.100037>

Re: mSystems00817-24R1 (Systematic identification of secondary bile acid production genes in global microbiome)

Dear Dr. Na Jiao:

Your manuscript has been accepted, and I am forwarding it to the ASM production staff for publication. Your paper will first be checked to make sure all elements meet the technical requirements. ASM staff will contact you if anything needs to be revised before copyediting and production can begin. Otherwise, you will be notified when your proofs are ready to be viewed.

Sincerely,
Naseer Sangwan
Editor
mSystems

Reviewer #1 (Comments for the Author):

The authors have addressed all of the previous comments and incorporate suggestions. Overall it is a nice manuscript, congrats.